# Recent advances in enantioselective ring-opening polymerization and copolymerization

Xiaoyu Xie[1,2], Ziyu Huo[1,2], Eungyo Jang[1] & Rong Tong [1✉]

Precisely controlling macromolecular stereochemistry and sequences is a powerful strategy for manipulating polymer properties. Controlled synthetic routes to prepare degradable polyester, polycarbonate, and polyether are of recent interest due to the need for sustainable materials as alternatives to petrochemical-based polyolefins. Enantioselective ring-opening polymerization and ring-opening copolymerization of racemic monomers offer access to stereoregular polymers, specifically enantiopure polymers that form stereocomplexes with improved physicochemical and mechanical properties. Here, we highlight the state-of-the-art of this polymerization chemistry that can produce microstructure-defined polymers. In particular, the structures and performances of various homogeneous enantioselective catalysts are presented. Trends and future challenges of such chemistry are discussed.

Asymmetric catalysts provide invaluable routes to enantiopure pharmaceutical agents and intermediates. On the other side, stereoselective polymerization catalysts are the most commercially viable means to prepare stereoregular polymers, that is, polymers having regularity of the configurations of adjacent stereocenters along the polymer chain. Stereoregular polymers are usually crystalline and have improved thermal and mechanical properties relative to those of atactic polymers[1]. Ever since the birth of polymer chemistry, polymer stereochemistry, or chirality, has been one of the focal points. In 1929, Staudinger and coworkers predicted a correlation between the polymer's stereochemistry and its physical properties[2]. In 1947, Schildknecht synthesized the first stereoregular polymers—isotactic poly(isobutyl vinyl ether)—and attributed its semicrystalline properties to the polymer's ordered tacticity[3]. In 1955, Natta and coworkers discovered the first isoselective catalysts to prepare stereoregular polypropylene, leading to the large-scale production of isotactic polypropylene (*it*-PP)[4]. Since then, synthesis of stereoregular polymers has been actively pursued in both industrial and academic laboratories[1,5–13].

Several methods to prepare isotactic polymers are shown in Fig. 1. In the first route, enantiopure monomers are polymerized to produce isotactic polymers (Fig. 1a). Nevertheless, obtaining enantiopure monomers in industry is challenging, requiring additional purification and separation steps. Alternatively, racemic monomers can be polymerized by isoselective catalysts to produce stereoblock copolymers (Fig. 1b, route i). In lieu of the isoselective polymerization, enantioselective catalysts often lead to the production of gradient stereoblock copolymers when the enantioselectivity is not exclusive (Fig. 1b, route ii). Highly exclusive enantiopure enantioselective catalysts can selectively enchain one enantiopure monomer to produce isotactic polymers, leaving the other enantiomer unreacted (Fig. 1b, route iii). Finally, using a pair of enantioselective catalysts (e.g., the racemic mixture of chiral enantioselective complexes in route iii) may enable the preparation of stereocomplex of two complementary isotactic polymers from racemic monomers (Fig. 1b, route iv). Notably, the interaction between the stereocomplex polymers results in altered physical properties in comparison to the parental polymers[14–19]. Compared with other routes, the enantioselective polymerization

---

[1]Department of Chemical Engineering, Virginia Polytechnic Institute and State University, 635 Prices Fork Road, Blacksburg, Virginia 24061, USA. [2]These authors contributed equally: Xiaoyu Xie, Ziyu Huo. ✉email: rtong@vt.edu

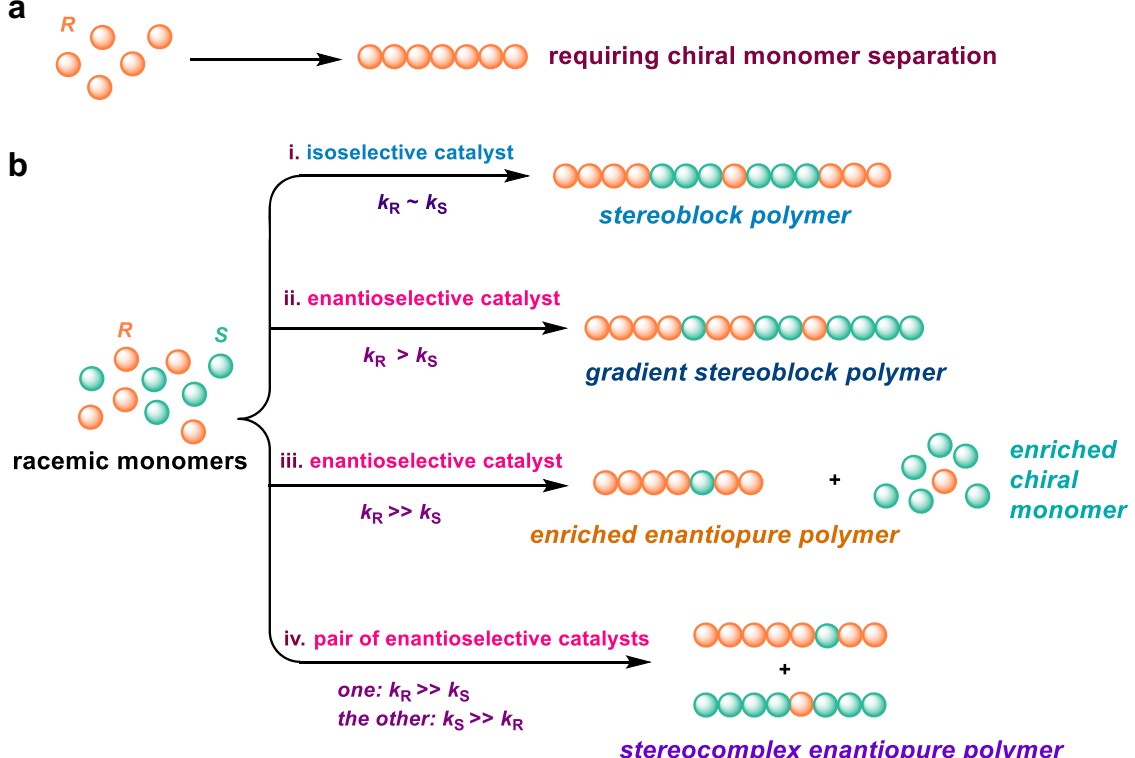

**Fig. 1 Stereoselective polymerization of racemic monomers. a** Synthesis of enantiopure polymers from purified enantiopure monomers, which requires additional steps for monomer separation. **b** Different polymerization strategies to prepare stereoregular copolymers using isoselective or enantioselective polymerization catalysts.

strategy (the last two routes) offers attractive means to produce stereoregular polymers such as stereocomplex enantiopure polymers with improved physical and mechanical properties.

Although there has been long history of producing stereoregular polymers, the development of enantioselective polymerization catalysts for degradable or recyclable polymers has recently undergone a surgency, aware of the excessive non-degradable plastic waste production[13,20–25]. Ring-opening polymerization (ROP) is a chain-growth polymerization strategy that encompasses many propagation mechanisms such as cationic, anionic, and coordination–insertion[26–29]. Many degradable polymers, including polyesters, polyethers, and polycarbonates, can be efficiently synthesized via ROPs or ring-opening copolymerizations (ROCOPs)[10,30,31]. This review focuses on recent advances in developing enantioselective catalysts in ROP and ROCOP to produce degradable polymers from racemic monomers. These monomers or monomer combinations mainly include lactide, lactone, epoxide, epoxide/CO$_2$, epoxide/anhydride, thiirane, and $N$-carboxyanhydride.

We highlight the novel synthetic routes and catalyst design strategies, and discuss the catalytic mechanisms in these enantioselective polymerizations. We organize this review based on the types of the monomers. We focus on the chemistry for preparing polymers having optical isomerisms in the polymer backbone, but the *cis-trans* (geometric) isomerism (e.g., polyisoprene) is not discussed. Because the emphasis is on enantioselective polymerization enabled by the catalyst or initiator, the polymerization chemistry of enantiopure monomers is not covered. Notably, stereoselective polymerization without chiral preferences on the monomers will not be focused on, although such systems are mentioned occasionally for comparison.

## Basic concepts in enantioselective polymerization

Stereocontrolled polymerization can be mediated by two distinct mechanisms, namely, chain-end control (CEC) or enantiomorphic site control (ESC)[1,32]. (Fig. 2) In the former case, the chirality of the chain-end monomer determines the chirality of the next monomer to be inserted, even if the catalyst contains a chiral component (Fig. 2a). In the ESC-mediated polymerization, the chirality of the catalyst determines the chirality of the next monomer unit. (Fig. 2b) The monomer enchainment mechanism can be easily identified by observing the stereochemical errors propagating in a polymer chain: stereoerrors can be propagated in the CEC-mediated polymerization; whereas in the ESC-mediated polymerization, correction of stereoerrors can occur because the chiral catalysts direct the stereochemical events, leading to an isolated stereoerror.

The NMR spectrometry is used to characterize the polymer stereochemistry. The Bovey formalism describes relationships between adjacent stereogenic centers (dyads) in the polymer tacticity, where an '*m*' for *meso* (same configuration), and an '*r*' for *racemic* (opposite configuration)[33]. For CEC-mediated enchainment, the parameters $P_m$ and $P_r$ refer to the probability of *meso* and *racemic* linkages ($P_m + P_r = 1$), respectively. For ESC-mediated enchainment, another parameter α—the site control selectivity—is often used to represent the probability of selecting one enantiomer of the monomer for enchainment. When α is 1 or 0, an isotactic polymer forms; an α of 0.5 results in an atactic polymer.

For the enantioselective polymerization, enantiopure catalysts can stereoselectively resolve racemic monomers to produce isotactic polymers enriched in one enantiomer of the monomer, thereby leaving unreacted monomer enriched in the other enantiomer (Fig. 1b, route iii). In most cases, such

**Fig. 2 Two distinct mechanisms of stereoselective polymerization. a** Chain-end control mechanism. **b** Enantiomorphic site control mechanism.

enantioselectivity is mediated by the ESC mechanism. The selectivity factor, often referred as the s factor (*s*), is the quantitative measurement of stereocontrol in such a system. The s factor should be calculated using the Eq. (1) based on the monomer conversion (*c*), which is often determined by NMR, and the enantiomeric excess (*ee(m)*) of the unreacted racemic monomer mixture, which can be determined by chiral HPLC. Alternatively, NMR spectrometry could be used to measure the enantiomeric purity of repeating units in the polymer (*ee(p)*) using ESC statistics[1,34].

$$s = \frac{\alpha}{1-\alpha} = \frac{\ln[(1-c)(1-ee(m))]}{\ln[(1-c)(1+ee(m))]} = \frac{\ln[1-c(1+ee(p))]}{\ln[1-c(1-ee(p))]} \quad (1)$$

For the ROP of lactide (LA) mediated by ESC, the site control selectivity (α), which can be calculated based on NMR spectra, can also be used to calculate the s factor, as shown in Eq. (1)[35]. Note that in the literature, the s factor also can be referred as the ratio of the rate constant, $k_{rel}$, for polymerization of the faster-reacting enantiomer to that of the slower-reacting enantiomer of the monomer, assuming that the reaction is first order in monomer. Nevertheless, the $k_{rel}$ value should not be calculated using individual enantiomer's kinetic rates in the absence of the other enantiopure monomer, because the polymerization rate of individual enantiomer in the racemic mixture is different from that of the pure enantiomer. To avoid confusion, we use s factors throughout the review unless the kinetic rates of the enantiomer is determined in the racemic monomer mixture (which is challenging as the isotope-labeled monomer should be used to differentiate the individual enantiomer in the racemic mixture)[36]. Notably, routine [13]C NMR spectroscopic experiments are now often used to reveal both the tacticity, the stereoregularity, and the degree of regioregularity[37,38], but such quantification information is often absent in the old literature.

### Enantioselective polymerization of lactide

Using different catalysts, the ROP of the racemic lactide (*rac*-LA, the mixture of D-LA ((*R,R*)-LA) and L-LA ((*S,S*)-LA)) can lead to isotactic stereoblock poly (lactic acid) (PLA) or heterotactic PLA, whereas *meso*-LA ((*R,S*)-LA) can produce heterotactic or syndiotactic PLA[35,39–42]. Here we focus on the enantioselective ROP of *rac*-LA. We discuss the Al catalysts first as they have been extensively applied in the enantioselective ROP of LA, followed by discussions about other metal complexes, and organocatalysts for such polymerization.

**Al-based complexes**. In 1996, Spassky and coworkers first reported using chiral (*R*)-**Al-1** complex for enantioselective ROP

of *rac*-LA (Fig. 3a)[43]. At 70 °C, the (*R*)-**Al-1** catalyst exhibits a 20:1 preference for the polymerization of D-LA over L-LA. At the LA conversion of less than 50%, the polymer microstructure is predominantly isotactic poly(D-LA). At high monomer conversions, the obtained polymer exhibits a tapered gradient microstructure, where the monomer composition varied from all (*R*)-units to all (*S*)-units (Fig. 3a). In 1999, Coates and coworkers showed that the (*R*)-**Al-2** preferentially attacked at the carbonyl group adjacent to the (*R*)-stereogenic atom in *meso*-LA, followed by the Al-binding to the (*S*)-lactic acid unit (Fig. 3b)[35,44,45]. To this end, the enantiopure (*R*)-**Al-2** polymerizes *meso*-LA to produce syndiotactic PLA via the ESC mechanism with an s factor of 49 (α = 0.98), while *rac*-**Al-2** polymerizes *meso*- or *rac*-LA to heterotactic or isotactic stereoblock PLA, respectively (Fig. 3b). Based on such chiral Al complex, Duda and coworkers developed a two-step polymerization of *rac*-LA by consecutive addition of homochiral (*S*)- and (*R*)-**Al-2** complexes into the polymerization mixture, resulting in the stereocomplexation of stereoblock PLAs with a high melting temperature ($T_m$) of 210 °C (Fig. 3c)[46]. Later, Coates, Meyer, and coworkers used (*R*)-**Al-2** to regioselectively polymerize (*S*)-methyl glycolide with the ring-opening exclusively at the glycolide acyl–oxygen bond site, as the chiral Al complex and the steric hindrance all favored the ring-opening on that specific site (Fig. 3d)[47]. On the other hand, using (*S*)-**Al-2** resulted in a moderate regioselectivity of 78%, presumably due to the mismatch between the chiral Al complex and (*S*)-LA at the chain end.

Additionally, in 2002, Feijen and coworkers reported that two chiral (*R,R*)-salcy-Al complexes (salcy, *N,N′*-bis(salicylidene) cyclohexanediimine), (*R,R*)-**Al-3** and (*R,R*)-**Al-4**[48,49], preferentially polymerized L-LA (Fig. 3e). The reported s factors were lower than the ratio between individual monomer's kinetic rates, suggesting stereoerrors occurred in the ROP of *rac*-LA. Very recently, Wang and coworkers studied the phenoxy substituent groups on the (*R,R*)-salcy ligands, and found that replacing *para*-[t]Bu with Br in (*R,R*)-**Al-3** (noted as (*R,R*)-**Al-5**) could increase the s factor to 4.6 at the conversion of 52%; while replacing the *ortho* [t]Bu group on the phenolate with [i]Pr (noted as (*R,R*)-**Al-6**) switched the monomer chirality preference from L-LA to D-LA with an s factor of 3.1 (Fig. 3e)[50].

In addition to monomeric (salen)Al complex (salen, *N,N′*-bis(salicylidene)ethylene diamine), in 2021, Zhang and coworkers prepared bimetallic Al (bi-Al) complexes based on chiral binaphthalene salen ligands ((*S*)-**Al-7**), and showed one of such bi-Al complexes could mediate enantioselective ROP with an s factor around 7 at the conversion ~50% with the preference on L-LA (Fig. 3f)[51]. The ROP of *rac*-LA by *rac*-**Al-7** complexes generated a gradient stereoblock PLA with a $M_n$ ($M_n$, number-average molecular weight) up to 60.7 kDa and a $P_m$ of 0.83.

**Fig. 3 Al complexes for enantioselective ring-opening polymerization (ROP) of lactide (LA). a** The enantioselective ROP of racemic-LA (*rac*-LA) by **Al-1** at low LA conversions. **b Al-2**-mediated regioselective ROP of *meso*-LA and *rac*-LA. **c** Sequential addition of **Al-2** enantiomers to prepare stereoblock PLA. **d Al-2**-mediated regioselective ROP of (*S*)-methyl glycolide. **e** The stereoselective polymerization of *rac*-LA to produce stereoblock PLA by **Al-3, Al-4, Al-5** or **Al-6**. **f** The enantioselective ROP of *rac*-LA by **Al-7** at low LA conversions. **g Al-8**-mediated ROP to synthesize gradient stereoblock PLA. **h** Proposed polymer exchange mechanism in **Al-9**-mediated ROP of *rac*-LA. **i Al-10**-mediated syndioselective ROP of *meso*-LA. **j** Proposed possible ring-opening reaction pathways of LA via the enantiomorphic site control mechanism.

with an s factor of 1.6 (Fig. 3g)[52]. Mechanistic studies suggested that the combination of ESC and CEC mechanisms might be attributed to the formation of gradient stereoblock PLA[52]. Kol and coworkers later identified a chloro-substituted racemic salan Al complex (**Al-9**) mediated heteroselective ROP of *rac*-LA ($P_r = 0.98$ at 50 °C), while its enantiopure Al complex had significantly lower hetereoselectivity ($P_r = 0.64$ at 70 °C, Fig. 3h). They proposed that after each LA insertion, the catalytic site was blocked until an exchange occurred with the opposite polymer chain bound to the opposite salan Al enantiomer[53,54]. The Kol group also reported the asymmetric chiral salan Al complex (**Al-10**) followed similar auto-inhibition/exchange mechanism in the ROP of *meso*-LA with a high syndiotacticity degree (α) of 0.96 (Fig. 3i)[54,55]. Nevertheless, the stereoregular PLA mediated by these chiral salan catalysts did not have high molecular weight (MW). More highly active chiral Al complexes are needed to prepare high-MW stereoregular PLAs from *rac*-LA.

Despite intensive catalyst studies, the detailed ESC mechanism for such Al-mediated enantioselective ROP of LA has not been fully understood. On the other hand, the ROP of LA via the CEC mechanism shows that the ROP of LA proceeds via the nucleophilic attack (TS1), followed by the ring-opening process for enchainment (TS2)[56]. In 2020, Talarico and coworkers re-examined the ESC mechanism of chiral-Al complex-mediated ROP of LA via the DFT computation (DFT, density function theory)[57,58]. They proposed that between the two transition states (TS1 and TS2), a chain-monomer exchange could exist (equivalent to the backslip in the Ziegler-Natta catalysis[59]), which may lead to changes in the ligand wrapping mode of the active site (Fig. 3j). The computation results could also agree with Kol and coworker's proposed chain-exchange mechanism during the ROP[54]. Such complicated ESC-mediated enantioselective ROP might add difficulties in rationally designing catalyst ligands to prepare exclusively enantioselective catalysts for PLA polymerization.

**Other metal complexes**. Besides Al complexes, other chiral metal complexes have been tested for enantioselective ROP of *rac*-LA. Ma et al. showed chiral aminophenolate **Zn-1** and **Mg-1** complexes exhibited ratios of $k(R,R)/k(s,s)$ around 3, and these complexes usually showed higher polymerization rates compared to Al complexes and produced stereoblock PLA (Fig. 4)[60–62]. Mehrkhodavandi et al. developed a chiral (*R,R*)-di-indium complex (*R,R*)-**In-1**, which showed a $k(s,s)/k(R,R)$ ratio of 5.1, and the ROP of *rac*-LA by such In complex led to a gradient stereoblock PLA with a $P_m$ of 0.77 (Fig. 4)[63]. Recently, Wang and coworkers developed a Zn complex with a chiral bis(oxazolinylphenyl) amido ligand (**Zn-2**) to mediate the ROP of *rac*-LA with an s factor around 4 at the LA conversion ~ 50%; the stereoblock PLA was obtained at high monomer conversions with a $P_m \sim 0.8$ (Fig. 4)[64].

Apart from salen ligands, other chiral ligands were explored to prepare Al complexes for the enantioselective ROP of *rac*-LA. In 2014, Lamberti, Kol, and coworkers developed chiral salalen Al complexes (**Al-8**) to prepare gradient stereoblock copolymers

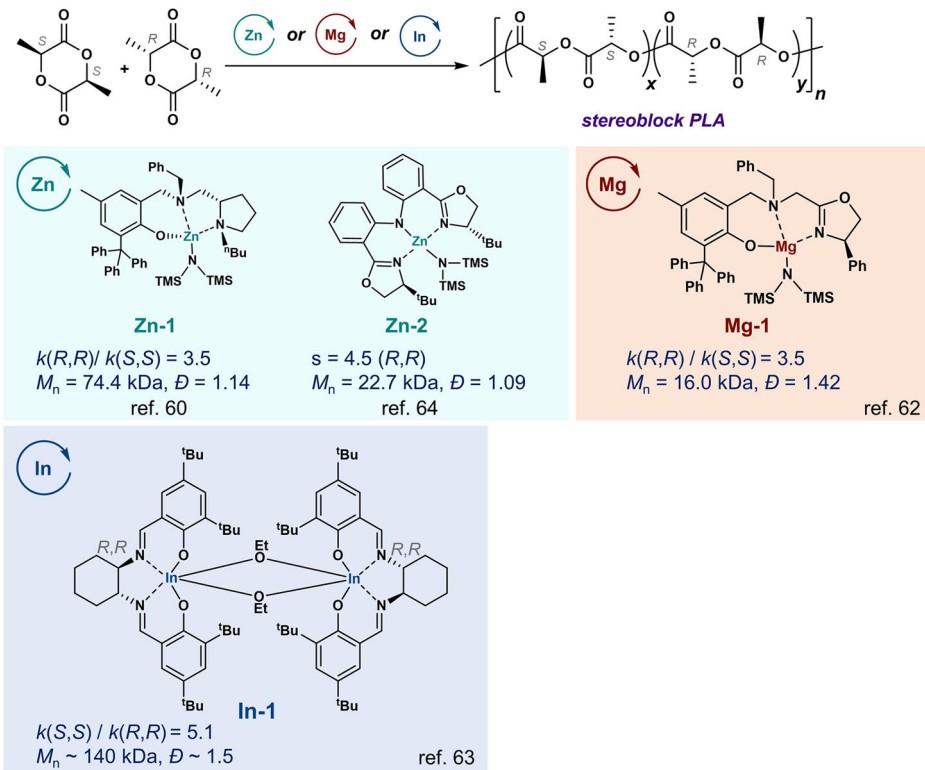

**Fig. 4 Zn, Mg, and In complexes for stereoselective ring-opening polymerization of racemic lactide.** The complexes lead to the production of gradient stereoblock poly(lactic acid), though the metal complexes show chiral preference on the monomers.

**Organocatalyst**. The Chen group first investigated the organocatalyst-mediated enantioselective ROP of *rac*-LA using cinchona alkaloids **O-1**[65]. The initial identified bifunctional catalyst, β-isocupreidine, consists of a chiral nucleophilic tertiary amine and an acidic phenol moiety (Fig. 5). The β-isocupreidine/benzyl alcohol-mediated ROP of *rac*-LA in dichloromethane afforded a gradient stereoblock PLA with a $P_m$ of 0.68 at room temperature. Kinetic studies showed a preferential ROP of L-LA using the combination of β-isocupreidine/benzyl alcohol, with an s factor of 4.4 at the *rac*-LA conversion of 48%. In 2015, the same group furnished the β-isocupreidine core with a thiourea-linked *R*-binaphthyl-amine, and both moieties were proposed to function as H-bond donors to exert stereo-differentiation (Fig. 5, **O-2**)[66]. Indeed, such catalyst with benzyl alcohol in *o*-difluorobenzene increased the s factor to 53 at the *rac*-LA conversion of 50%. Meanwhile, Satoh and coworkers also reported chiral BINOL-derived phosphoric acids **O-3** (BINOL, 1,1′-bi-2-naphthol) could mediate enantioselective ROP of *rac*-LA[67]. For the (*R*)-phosphoric acid-mediated ROP of *rac*-LA at 75 °C in toluene using 3-phenyl-1-propanol as an initiator, D-LA was preferred, and the s factor was found as 28.3 at 49.0% conversion of *rac*-LA.

In 2017, Mecerreyes, Cossío and coworkers showed that diastereomeric *N*-methylated prolines **O-4** could mediate enantioselective ROP of *rac*-LA in dichloromethane at room temperature (Fig. 5)[68]. Specifically, for the ROP of *rac*-LA in the presence of cocatalyst 1,8-diazabicyclo[5.4.0]undec-7-ene (DBU) with the monomer conversion ∼ 50%, *endo*-proline promoted the ROP of D-LA, whereas *exo*-proline preferentially polymerized L-LA. Notably, the ROP of *rac*-LA in the presence of a mixture of *exo*- and *endo*-prolines resulted in atactic PLA ($P_m = 0.60$). DFT studies suggested the high density of chiral and prochiral centers in the organocatalysts may contribute to the selectivity of chiral LA monomers.

Moreover, other organocatalysts, such as chiral thioureas, were explored for enantioselective ROP of *rac*-LA. In 2018, Taton, Coulembier, Dove and coworkers found that the chiral Takemoto's thiourea **O-5** (Fig. 5) had a moderate s factor around 3.4 for the ROP of *rac*-LA in toluene to generate a gradient stereoblock PLA with a $P_m$ of 0.88 at room temperature[69]. The addition of a phosphazene base into such system could markedly improve the polymerization rate to produce a gradient stereoblock PLA with a $P_m$ of 0.96[70]. In 2020, Wang and coworkers also showed the iminophosphorane–thiourea catalysts **O-6** (Fig. 5) had a $k(s,s)/k(R,R)$ ratio of 1.3 and an s factor of 1.4 for the ROP of *rac*-LA using benzyl alcohol as an initiator in dichloromethane at room temperature[71]. Nevertheless, all organocatalyst-mediated enantioselective ROP of LA could not produce stereoregular PLAs with MWs higher than 20 kDa.

## Enantioselective polymerization of β-lactone or 8-membered diolide

Polyhydroxyalkanoates (PHAs), including poly-3-hydroxybutyrate (P3HB), is a type of highly crystalline thermoplastic polyesters produced by bacteria and other microorganisms[72,73]. Isotactic P3HB exhibited PP-like $T_m$, (160 − 170 °C), and excellent barrier properties superior to PE and PET[74]. However, P3HB is thermally unstable with a relatively low degradation temperature close to $T_m$ (∼ 250 °C)[75], and is extremely brittle with a fracture strain (ε) of 4%[76]. Here we focus on the enantioselective ROP from β-lactone or diolide to produce stereoregular PHAs.

**β-lactone**. In 1980s, Spassky and coworkers found that the complex of ZnEt₂ and (*R*)-3,3-dimethyl-1,2 butanediol (DMBD) preferentially ring-opened (*R*)-β-butyrolactone (*R*-BL) with a $k(R)/k(S)$ ratio of 1.6, resulting in the mixture of isotactic P3HB

**Fig. 5 Organocatalysts for enantioselective ring-opening polymerization of racemic lactide (*rac*-LA).** Note that s = 4.4 (*S*, *S*) means that the selectivity factor to (*S*, *S*)-LA is 4.4.

and amorphous copolymers (Fig. 6a)[77]. Using CdMe$_2$ and *R*-DMBD resulted in slightly preferential ROP on *R*-BL. The complex between ZnEt$_2$ and (*R*)-DMBD also exhibited moderate stereoselectivities for *R*-α-methyl-α-ethyl-β-propiolactones (α-Me-β-PL, s = 1.05)[78], and *R*-α-methyl-α-*n*-propyl-β-propiolactone (s = 1.25)[79]. Meanwhile, Takayama and coworkers found the (*R*,*R*)-(salcy)Co$^{II}$ complex (**Co-1**) and AlEt$_3$ could preferentially polymerize *R*-BL, without detailed measurement of selectivity factors (Fig. 6b)[80]. Later, though various discrete metal complexes were reported for stereoselective ROP of *R*-BL, many of them exhibited excellent syndioselectivities (*P*$_r$ > 0.9), instead of high isoselectivities and enantioselectivies[81–83]. Very recently, Rieger and coworkers' in-situ-generated catalyst based on Y[N(SiHMe$_2$)$_2$]$_3$(THF)$_2$ and a racemic salan ligand could lead to isoselective ROP of *rac*-BL to produce stereocomplex P3HB with a *P*$_m$ of 0.89 (no stereoblock microstructures on NMR), resulting in strong and ductile copolymers whose tensile properties outperform bacterial-synthesized P3HB (Fig. 6c)[84].

In addition to metal complexes, enzymes have been investigated for the enantioselective ROP of β-lactones. In 1996, the Gross group used lipase PS-30 to prepare *S*-enriched poly(α-Me-β-PL) in toluene with *M*$_n$s around 2.9 kDa: at 47% conversion of the racemic monomers, the s factor was 4.7, and the unreacted α-Me-β-PL having 45% *ee*. Wang et al. identified lipase ESL-001 to preferably polymerize *R*-BL from racemic BLs[85]. In 2000, Kobayashi and coworkers found the lipase-mediated copolymerization of racemic β-BL with achiral 12-dodecanolide in diisopropyl ether could give *S*-enriched copolymers with 76% *ee* of β-BL in the copolymer at 35% monomer conversion[86]. Nevertheless, most enzyme-mediated enantioselective ROP of β-lactones could only produce low-MW polymers with MWs less than 20 kDa.

**8-membered diolide**. To overcome the low reactivities in the enantioselective ROP of β-lactones and produce high-MW stereoregular PHAs, in 2018, Chen and Tang developed another synthetic route to prepare PHAs with high enantioselectivities[87]. Instead of focusing on the four-membered β-lactone, they prepared a group of eight-membered diolide monomers from biosourced dimethyl succinate in four steps. They used enantiopure

chiral yttrium complex (**Y-1**) having bulky salcy ligands to mediate enantioselective ROP of such diolide, and the enantiomerically pure **Y-1** complex (either (*R*, *R*) or (*S*, *S*) enantiomer) preferentially polymerized one enantiomer of diolide to yield optically pure (*R*) or (*S*)- P3HB (*P*$_m$ = 0.99), and mechanistic studies suggested ESC-mediated mechanism attributed to the high enantioselectivity (Fig. 6d). Using *rac*-**Y-1** complex could produce isotactic stereocomplex P3HB with a high isotacticity of up to 99%, a high *M*$_n$ up to 154 kDa, and a *T*$_m$ of 175 °C[87]. Notably, the *rac*-**Y-1** complex could polymerize *meso*-diolide to generate syndiotactic (*st*) P3HB with a *P*$_r$ of 0.81 and a *T*$_m$ of 141 °C (Fig. 6d)[88].

Notably, the synthesis of diolide from dimethyl succinate yields a mixture of *rac*- and *meso*- diastereomers (Fig. 6e). The enantioselectivity of **Y-1** enables the copolymerization of the mixture of diolide diastereomers to produce stereoregular P3HBs via kinetic resolutions[88]. Indeed, the Chen group showed the enantioselective ROP of *rac*-diolide (*rac*-DL$_{Me}$) and *meso*-DL$_{Me}$ could be achieved in a one-pot fashion at room temperature (Fig. 6e): the *rac*-diolide was consumed first, at which point 24% *meso*-diolide was also polymerized; this was followed by subsequent ROP of *meso*-diolide to provide a stereocomplex of tapered (*it*-P3HB)-*sb*-(*st*-P3HB) with improved ductility compared to isotactic *it*-P3HB[88]. Replacing the *rac*-DL$_{Me}$ with *rac*-DL$_{Et}$ in such one-pot copolymerization using less bulky **Y-2** could generate another stereocomplex of (*st*-P3HB)-*sb*-(*it*-P3HV) (P3HV, poly(3-hydroxyvalerate)); ROP rates of *meso*-DL$_{Me}$ more rapid than that of *rac*-DL$_{Et}$) with enhanced ductility and strength compared to either isotactic *it*-P3HB or *it*-P3HV (Fig. 6f)[88]. To this end, the Chen group performed one-pot copolymerization of all six DL$_{Me}$ and DL$_{Et}$ diastereomers using racemic **La-1** to produce gradient stereocomplex copolymers (*it*-P3HB)-*sb*-(*it*-P3HV)-*sb*-(st-P3HB)-*sb*-(*st*-P3HV), because of the monomer reactivities where *rac*-DL$_{Me}$ > *rac*-DL$_{Et}$ and *meso*-DL$_{Me}$ > *meso*-DL$_{Et}$ (Fig. 6g). Very recently, the same group applied such enantioselective ROP for asymmetric diolide, as the **Y-1** complex preferentially ring-opened the diolide at less hindered site, and thereby generating stereocomplex of isotactic PHAs with alternating pendant side-chain functional groups (Fig. 6h)[89].

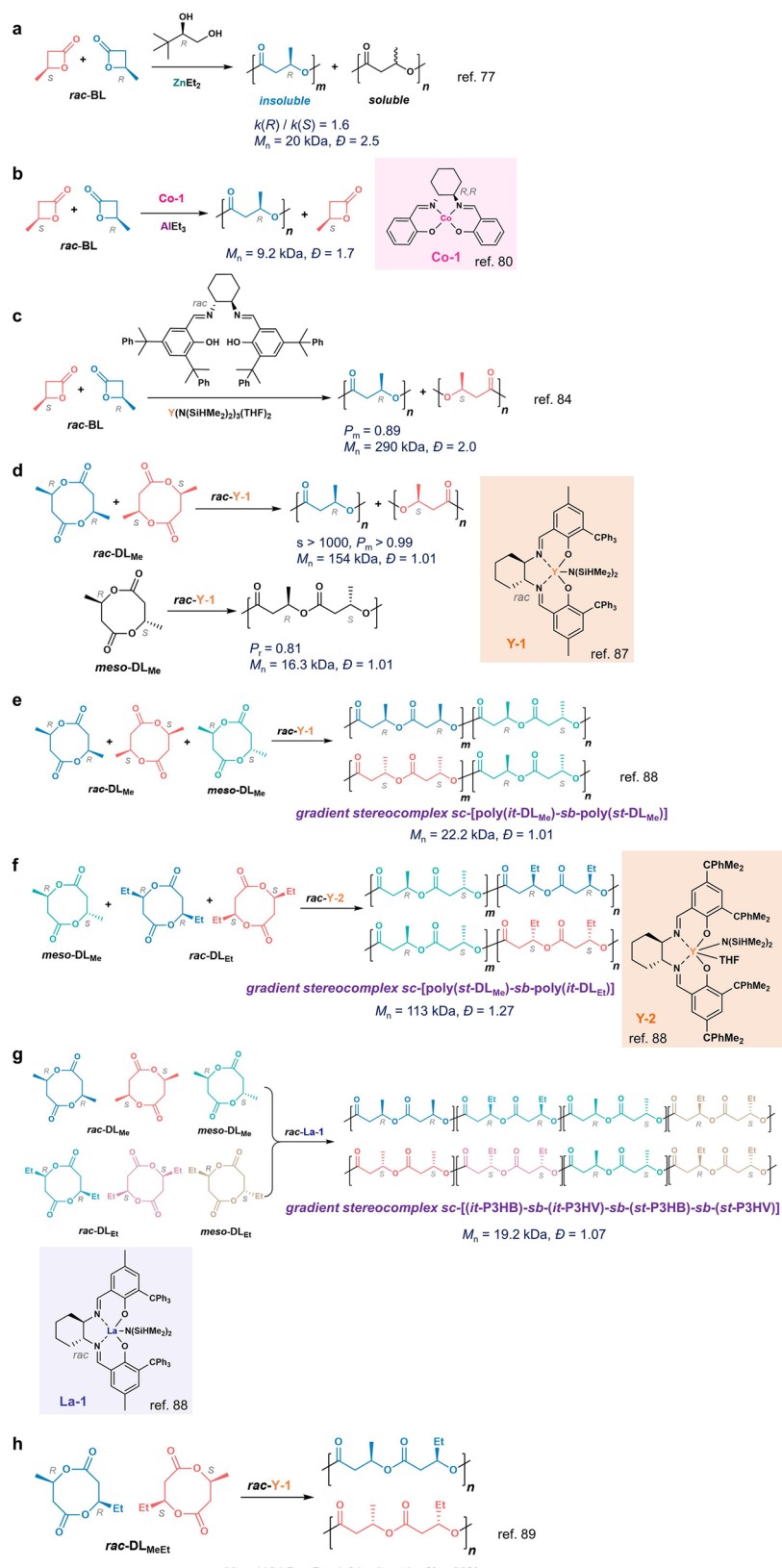

**Fig. 6 Enantioselective ring-opening polymerization (ROP) of racemic β-lactone (*rac*-BL, a-c) and racemic/meso 8-membered diolides (*rac*-/*meso*-DL, d-h).** Enantioselective ROP of *rac*-BL by **a** ZnEt$_2$ and (*R*)-3,3-dimethyl-1,2 butanediol, **b** **Co-1**/AlEt$_3$, and **c** the mixture of the salen ligand /Y complex. **d** Enantioselective ROP of *rac*-DL$_{Me}$ or *meso*-DL$_{Me}$ by *rac*-**Y-1**. **e** Stereosequence controlled polymerization of the mixture of *rac*-DL$_{Me}$ and *meso*-DL$_{Me}$ by *rac*-**Y-1**. **f** Stereosequence controlled polymerization of the mixture of *meso*-DL$_{Me}$ and *rac*-DL$_{Et}$ by *rac*-**Y-2**. **g** Stereosequence controlled polymerization of the mixture of *rac*-DL$_{Me}$, *rac*-DL$_{Et}$, *meso*-DL$_{Me}$ and *meso*-DL$_{Et}$ by *rac*-**La-1**. **h** Enantioselective and regioselective ROP of *rac*-DL$_{MeEt}$ by *rac*-**Y-1**.

## Enantioselective polymerization of other lactones

**ε-Caprolactone.** Poly(ε-caprolactone) (PCL) and its derivatives have drawn considerable attentions as PCL can be used as biocompatible materials for medical applications[90]. Two types of chiral monomers, 6- and 4-alkyl-ε-CL, have been explored for the enantioselective ROP. In 2002, Bisht et al. found the Novozym 435 could enantioselectively polymerize *rac*-4-Me-ε-CL and *rac*-4-Et-ε-CL at 60 °C, preferably polymerizing (S)-enantiomers at low monomer conversions[91]. Meijer and coworkers then determined that the s factors at 45 °C for 4-Me-ε-CL, 4-Et-ε-CL, and 4-*n*Pr-ε-CL were 16.9, 7.1 and 2.0, respectively, in which the enantiomer selectivity changed from (S)- to (R)-enantiomer in the polymerization of 4-*n*Pr-ε-CL (Fig. 7a)[92].

In contrast to the ROP of *rac*-4-Me-ε-CL, Novozym 435 could not mediate the ROP of *rac*-6-Me-ε-CL. In 2005, Meijer et al. found that the ring-opening reaction of 6-Me-ε-CL using Novozym 435 did occur, and such reaction was (S)-selective; however, after the ring-opening, the (S)-configured terminal secondary alcohol could not efficiently engage with the enzyme enchainment[93]. To overcome such problem, Meijer et al. identified a Ru complex (the mixture of RuCl₂(cymene) and 2-phenyl-2-aminopropionamide, Fig. 7b) to in situ racemize the terminal (S)-alcohol, which resulted in the active (R)-alcohol chain end for propagation. Such iterative tandem catalysis leads to (R)-enriched 6-Me-ε-CL oligomers (degree of polymerization < 5) with 100% monomer conversion and optical purity of up to 95% *ee*.

In 2007, Meijer and coworkers reported a switch from S- to R-selectivity from small to medium-sized lactone (ring sizes ≤7) to large-sized lactone (ring sizes ≥8) in the ROP of racemic ω-methylated lactones mediated by Novozyme 435[94]. The ROPs of racemic 7-MeHL, 8-MeOL, and 12-MeDDL all preferred (R)-monomer (Fig. 7c), yielding enantiopure (R)-polyesters with optical purity of up to 99% *ee*. The conformational change of lactones from *cis*-oid in small-sized lactones to *trans*-oid in large lactones might be attributed to such an enantioselectivity switch.

In addition to enzymes for enantioselective ROP of ε-caprolactones, metal complexes were explored for such polymerizations. In 2006, Feijen group found the chiral salen-Al complex could mediate enantioselective ROP of *rac*-6-Me-ε-CL (Fig. 7d)[95]. The chiral (R,R)-**Al-3** complex exhibited a preference for (R)-6-Me-ε-CL, and at 80% monomer conversion, the *ee* value was 50%, and the s factor was 1.63. On the other hand, no enantiomeric preference was found for chiral Al complex to polymerize *rac*-4-Me-ε-CL.

Recently, organocatalyst has been investigated for the enantioselective ROP of substituted ε-CL. In 2020, Wang and coworkers found that the chiral phosphoric acid could catalyze the enantioselective ROP of racemic 2-Bn-ε-CL and 6-Bn-ε-CL at 90 °C, with moderate s factors that prefer (R)-monomers around 2 at 50% monomer conversions, and eventually provided gradient stereoblock polymers (Fig. 7e, f)[96]. The chiral HPLC analysis showed that (R)-substituted-ε-CL monomers were preferentially polymerized when employing (R)-phosphoric acid (Fig. 7e), whereas (S)-substituted-ε-CL monomers were preferentially consumed when using (S)-phosphoric acid as the catalyst (Fig. 7f). The same group also evaluated chiral phosphoric acid for the enantioselective ROP of racemic 6-aryl-ε-CL[97] and 6-Me-ε-CL;[98] and the former had an s factor of 4.1 at the monomer conversion of 44% (Fig. 7f), while the latter showed an s factor of 7.3 at conversion of monomer conversion 44% (Fig. 7g). In all cases, gradient stereoblock copolymers were obtained when all monomers were consumed.

**Thiolactone.** Compared to the ester bond, the thioester bond is more reactive, and recent studies showed that polythioesters

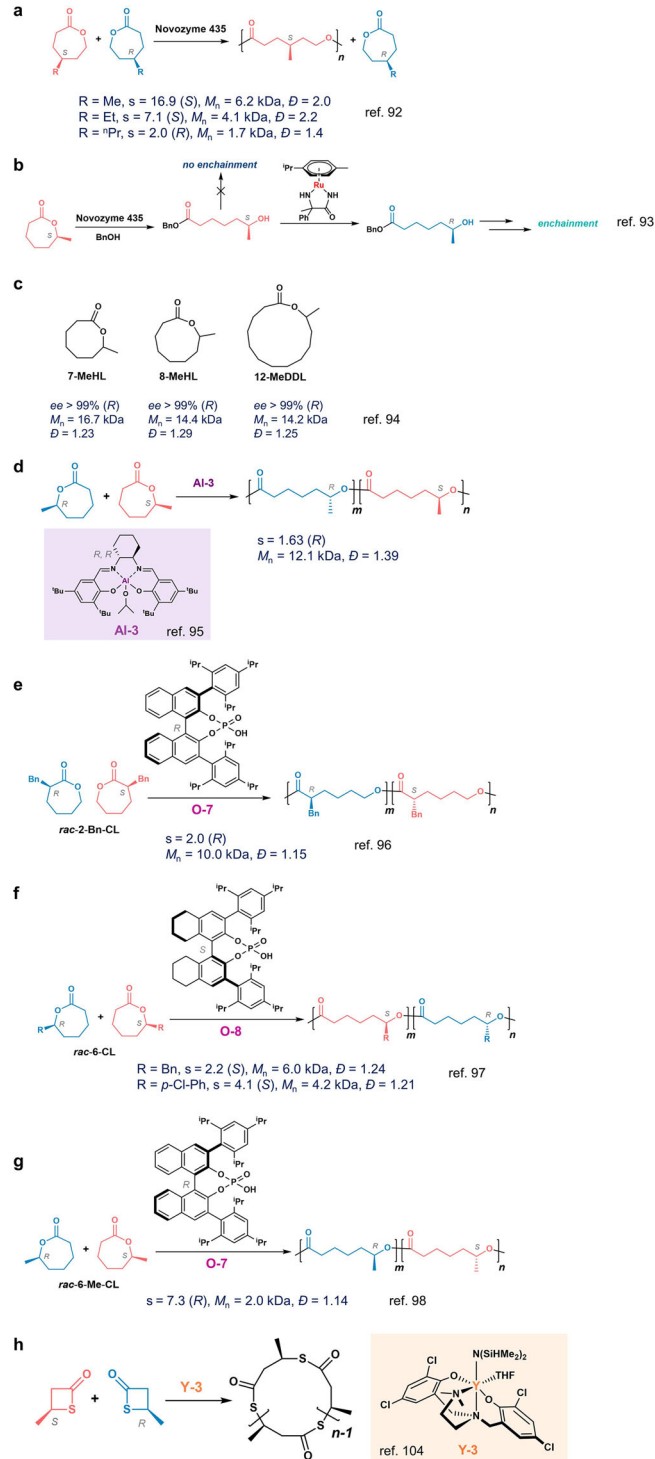

**Fig. 7 Enantioselective ROP of racemic ε-caprolactone (ε-CL, a–g) and racemic β-thiolactone (h).** **a** Enantioselective ROP of 4-alkyl-ε-CL by Novozyme 435. **b** Racemization of (S)-6-Me-ε-CL by the Ru complex to promote the enchainment of (R)-enriched CL. **c** The switch of chirality preference in Novozyme 435-mediated ROP of medium to large-sized lactones. **d** (R,R)-**Al-3**-mediated enantioselective ROP of *rac*-6-Me-ε-CL. **e** (R)-**O-7**-mediated ROP of enantioselective ROP of *rac*-2-Bn-ε-CL. **f** (S)-**O-8**-mediated ROP of enantioselective ROP of *rac*-6-CL. **g** (R)-**O-7**-mediated ROP of enantioselective ROP of *rac*-6-Me-ε-CL. **h** **Y-3**-mediated ROP of stereoselective ROP of *rac*-β-thiolactone.

could be degraded back to corresponding thiolactones[99–102]. Most of the ROP of racemic thiolactones reported so far used organocatalysts and produced atactic copolymers. Poly(3-thiobutyrate) (P3TB) with high MW (~175 kDa) was first prepared from a recombinant strain of *E.coli* cultured with racemic thiobutyric acids and was presumably highly isotactic[103]. In 2022, Carpentier, Guillaume, and coworkers developed tetradentate aminobis(phenolate) yttrium complexes to mediate either cyclic syndio- or enantioselective cyclic polymer stereocomplex[104]. The ROP of *rac*-TBL by the **Y-3** complex with less bulky substituents proceeded rapidly to produce isotactic cyclic P3TB with a $P_m$ of 0.90 (Fig. 7h). Notably, the stereocomplex of cyclic polymers were proposed, though kinetic studies and the *ee* measurement were not performed to eliminate the possibility of forming stereoblock microstructures via the chain-end control mechanism.

### Enantioselective polymerization of epoxides

Isotactic polyethers, e.g., isotactic polypropylene oxide (*it*-PPO), are semi-crystalline thermoplastics. Compared with polyolefins, polyethers have revealed low environmental existence: atactic PPO (*a*-PPO) can be metabolized by microorganisms through oxidation[105], and polyethers are susceptible to photo-oxidative degradation[106,107]. Commercial production of *a*-PPO uses double-metal cyanide catalysts in conjunction with alcohol chain transfer agents to produce low-MW polymers without tacticity control[108]. Though the synthesis of *it*-PPO was independently explored early in 1950s by Dow chemicals, Natta, and Price et al.[109–111], the large production of *it*-PPO from racemic propylene oxide (PO) has not been realized due to the low reactivities and stereoselectivity ratios[10]. Notably, in addition to enantioselectivity, regioselectivity plays an important role affecting the polyether microstructures for monosubstituted epoxide, i.e., head-to-tail, head-to-head, tail-to-tail linkages (Fig. 8a). For monosubstituted epoxides such as PO, enchainment can occur either at the methylene to give a secondary metal alkoxide, or at the methine with inversion to give a primary metal alkoxide (Fig. 8a). The resulted polyether is regio-regular when only one process dominates; and the polymer becomes regio-irregular when both processes occur, which give head-to-head and tail-to-tail linkages that can be characterized by NMR.

In 2005, the Coates group first reported enantioselective Co-catalyst **Co-2** capable of polymerizing *rac*-PO to exclusively regioregular stereocomplex *it*-PPO with over 99% triads and $M_n$s over 287 kDa[112]. Mechanistic studies suggest the bimetallic mechanism for PO enchainment: Co centers are separated by 7.1 Å, and an empty Co coordination site is allocated for the PO insertion into the polymer that is bound to the other Co site[113]. The proposed bimetallic mechanism led to the development of a series of chiral bimetallic complexes by the Coates group. In 2008, Coates and coworkers reported the first bimetallic enantioselective Co complex for the ROP of various monosubstituted epoxides, including alkyl, aryl, unsaturated, and fluorinated epoxides as well as glycidyl ethers (Fig. 8b)[114]. Combined with an cocatalyst salt, e.g., bis(triphenylphosphine)iminium acetate ([PPN][OAc]), the bimetallic **Co-3** complex could kinetically resolve various racemic epoxides into valuable enantiopure epoxides and isotactic polyethers with high $k_{rel}$ values (>200 for many epoxides), high *mm*-triad content (>98% for many epoxides), and high TOFs (up to 30,000 h$^{-1}$)[114,115]. In addition, the use of the racemic bimetallic complex allows the preparation of isotactic polyethers in quantitative yields (Fig. 8c)[115]. DFT computation studies reveal that the active form of the catalyst has two active *exo* anionic ligands (chloride or carboxylate) and an *endo* polymer alkoxide which can ring-open an adjacent cobalt-coordinated epoxide; additionally, the initiation is favored by an *endo* chloride ligand,

while propagation is favored by the presence of two *exo* carboxylate ligands[116]. Notably, the chirality of the diamines has little effect on this class of catalysts when the chirality of the binaphthol linker in the bi-Co complex was fixed[117]. Switching from a chiral binaphthol linker to an achiral biphenol linker (**Co-4**) resulted in a bi-Co complex, in which the stereochemistry of the diamines determined the preferred linker conformation[116].

Nevertheless, attempts to prepare *rac*-**Co-3** from racemic starting materials produced inseparable diastereomers, which impedes the preparation of stereocomplex polyethers. Coates and coworkers found that replacing the chiral diamines with achiral ethylene diamine in the bi-Co complex (**Co-5**) could also mediate enantioselective ROP of various monosubstituted epoxides to produce stereocomplex copolymers with *mm*-triad over 97% when combined with the cocatalyst [PPN][OPiv] (OPiv = pivalate; Fig. 8b)[117].

To prepare hydroxy-telechelic isotactic PPO—that is used industrially as a midsegment in polyurethane, in 2017, the Coates group developed one-pot synthesis of bimetallic salenen ligand for Cr complexes (**Cr-1**), which could tolerate di-alcohol chain-transfer agents in combination with [PNN]Cl; and (*S*)-**Cr-1** preferentially polymerized (*S*)-PO with an s factor ~ 60 (Fig. 8d)[118]. Notably, the produced UV light-degradable (*S*)-*it*-PPO exhibited dramatic strain hardening with an ultimate fracture strength comparable to that of Nylon-6,6 and an ultimate fracture strain similar to that of *it*-PP[119]. To increase the polyether's MW control and improve MW distributions, the same group introduced flexible inter between the active salen-Cr units to allow the complex to render conformations for cooperative ROP. They found the bi-Cr complex with the linker length of six methylenes (**Cr-2**) showed an optimal performance in terms of activity and selectivity, with an s factor of 59 in the ROP of *rac*-PO (Fig. 8d)[120].

### Enantioselective alternating copolymerization of epoxide/CO₂

The ever-increasing emissions of $CO_2$ as waste material from fossil-based industries pose chemists a grand challenge to efficiently transform $CO_2$ to value-added chemicals[121,122]. In contrast to many reactions requiring expensive stoichiometric reagents and having scale-up problems[123], the ring-opening copolymerization (ROCOP) of $CO_2$ and other monomers, e.g., epoxides, is well suited for large-scale deployment because it enables significant $CO_2$ sequestration, and produces valuable polymers with properties to replace existing petrochemicals[124].

The ROCOP of epoxide/$CO_2$ can be traced back to 1969[125,126], and such polymer chemistry has witnessed a renaissance in the last 20 years due to the development of new catalysts, which has been extensively reviewed by many others[10,12,30,127–132]. Here we focus on highlighting enantioselective ROCOP of epoxide/$CO_2$. Similar to the homopolymerization of epoxide, in addition to enantioselectivity, the regioselectivity of ROCOP of epoxide/$CO_2$ plays an important role in copolymer's microstructure.

**Monosubstituted epoxide/CO₂.** In 2003, Coates and coworkers first developed chiral (salcy)Co$^{III}$-based catalysts for the ROCOP of *rac*-PO and $CO_2$ to produce poly(propylene carbonate) (PPC) with over 90% head-to-tail linkages when combined with PPN salts[133,134]. The (*R,R*)-**Co-6** preferably polymerized (*S*)-PO over (*R*)-PO with an s factor of 2.8 (Fig. 9a, b)[133]. The Lu group found that replacing the axial group in chiral (salcy)Co$^{III}$X from acetate to binitrophenol (**Co-7**) and using cocatalyst $^n$Bu₄NCl or 7-methyl-1,5,7-triazabicyclo[4.4.0]dec-5-ene resulted in the increase of the s factor to 3.5[135], and 5.9[136], respectively. Coates et al. later found out the (*R,R*)-**Co-8** /[PPN]Cl catalyst system had an s factor of 9.7 for the enchainment of (*S*)- over (*R*)-PO at −20 °C,

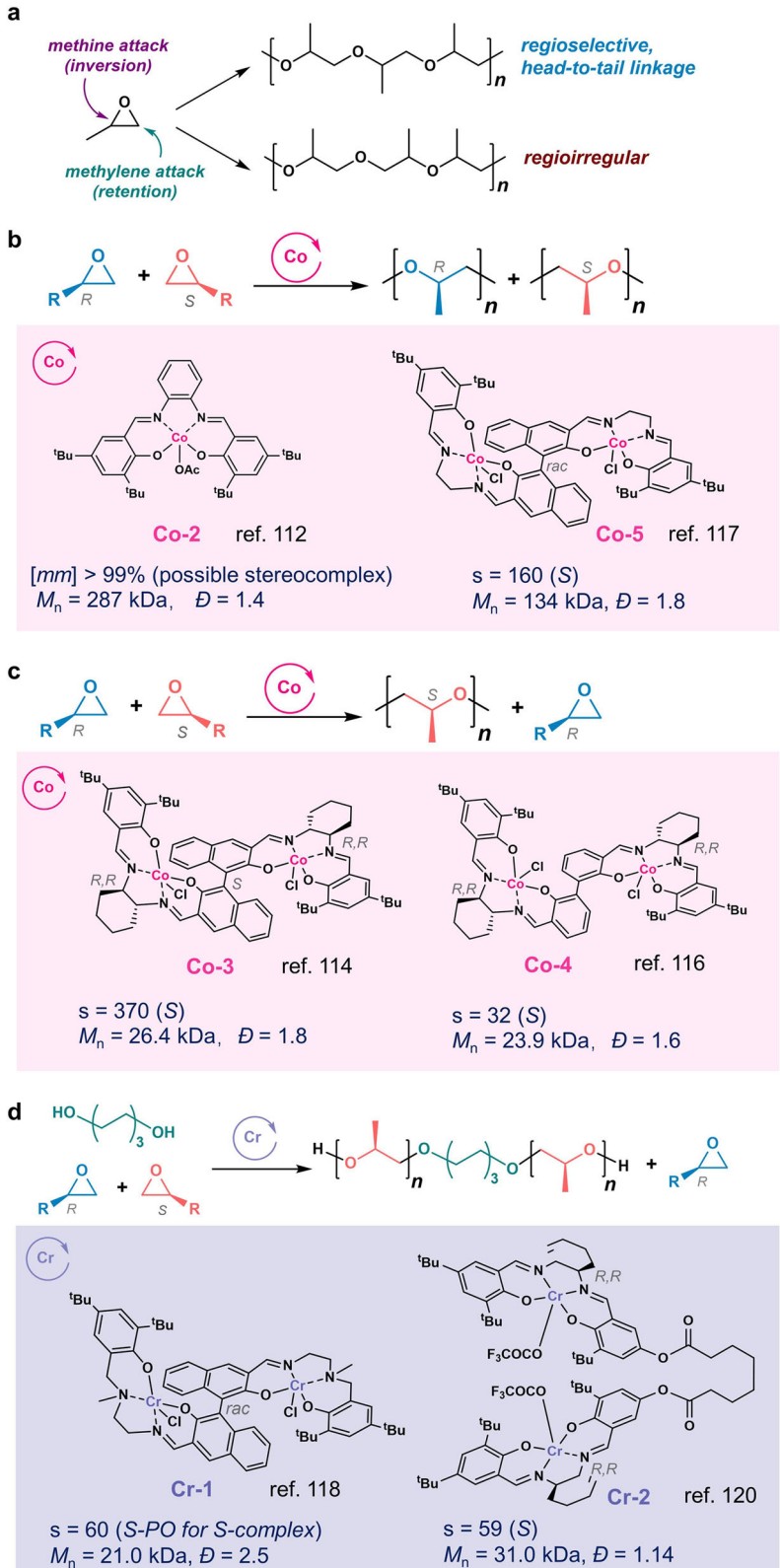

**Fig. 8 Enantioselective ring-opening polymerization (ROP) of racemic monosubstituted epoxide. a** Regio-selective ring-opening polymerization of propylene oxide (PO) to produce regioregular poly(propylene oxide) (PPO). **b Co-2** or **Co-5** mediated enantioselective ROP of *rac*-PO to produce stereocomplex PPO. **c Co-3** or **Co-4** mediated enantioselective ROP of *rac*-PO to produce (*S*)-PPO. **d** Isoselective ROP of *rac*-PO with the diol as the chain-shuttling agent to produce isotactic PPO.

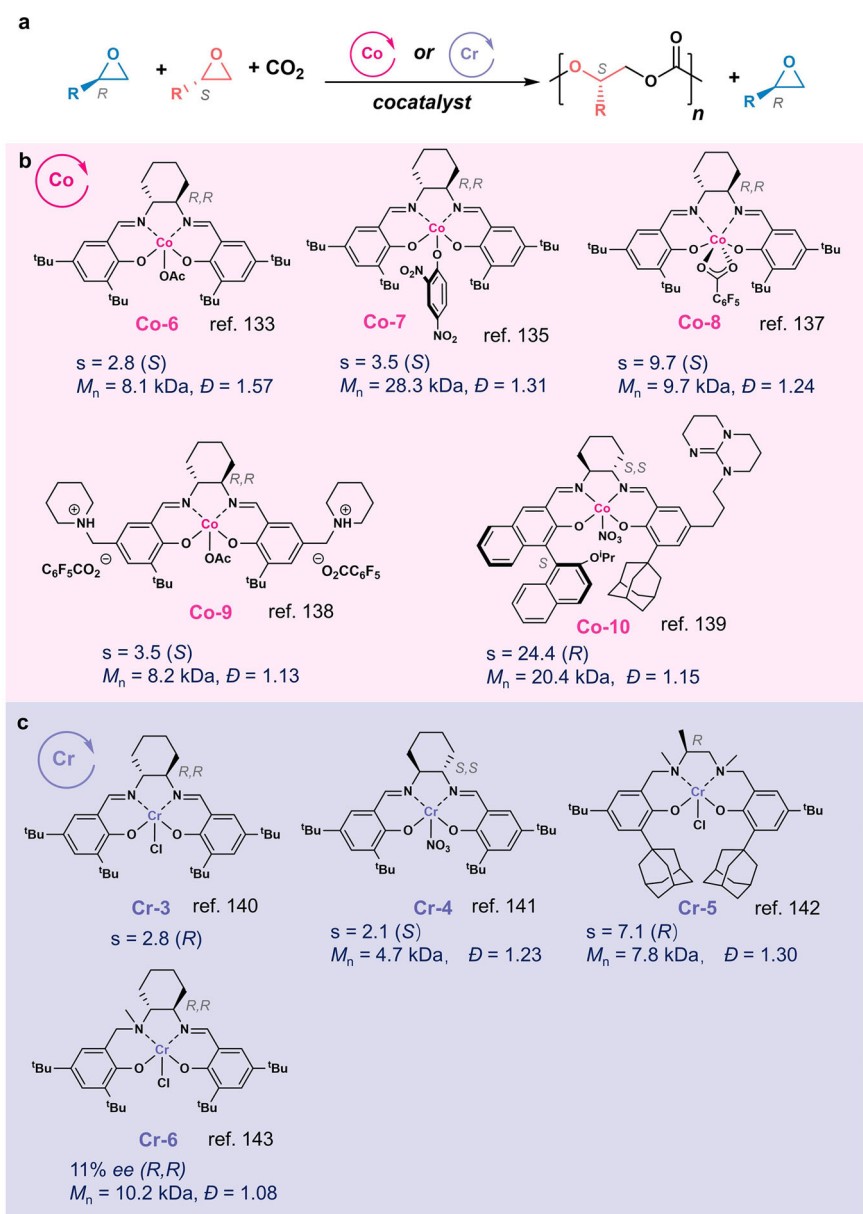

**Fig. 9 Enantioselective ring-opening copolymerization (ROCOP) of racemic monosubstituted epoxide/CO₂. a** The enantioselective ROCOP reaction scheme. **b** Co and **c** Cr complexes used in the polymerization in **a**.

resulting in a near perfectly regioregular PPC with 98% head-to-tail connectivity[137]. Nozaki et al. incorporated cocatalyst onto the salen ligand (**Co-9**) to improve the ROCOP reactivities, and produced gradient stereoblock PPC with the head-to-tail ratio of 95% and an s factor value of 3.5[138]. Such gradient stereoblock PPC showed significantly increased degradation temperature compared with stereocomplex or enantiopure PPCs. Along this line, the Lu group developed an asymmetrical salen ligand containing a chiral BINOL and an appended base 1,5,7-triabicyclo[4.4.0] dec-5-ene for the Co complex (**Co-10**), which mediated ROCOP of *rac*-PO/CO₂ with a high s factor of 24.4 at −20 °C[139].

Cr-based catalysts have been explored for enantioselective ROCOP (Fig. 9c). In 2000, Jacobsen et al. reported enantioselective copolymerization of CO₂/1-hexene oxide catalyzed by chiral (*R,R*)-salen-Cr catalyst (**Cr-3**) with an s factor of 2.8[140]. Later, the Lu group and the Nozaki group individually reported chiral (salen)Cr (**Cr-4**)[141], (salan)Cr (**Cr-5**)[142], and (salalen)Cr (**Cr-6**)[143] for the copolymerization of PO/CO₂, with s factors ranging from 2 to 7.

***Meso*-epoxide/CO₂.** The opposite configurations in the *meso*-epoxides allow for the synthesis of isotactic (−RR− or −SS−) and syndiotactic (−RR−SS−) polycarbonates (Fig. 10a). In 1999, the Nozaki group first reported the mixture of ZnEt₂ and (*S*)-α,α-diphenyl(pyrrolidin-2-yl)methanol could mediate ROCOP of CO₂ and cyclohexene oxide (CHO) to produce perfectly alternating poly(cyclohexene carbonate) (PCHC, Fig. 10b)[144]. The subsequent hydrolysis of the obtained PCHC showed the polymer has 73% *ee*, and no *cis*-diol was found in the degradation, suggesting the ring-opening reaction proceeds through an S_N2-type mechanism. The same group then treated such Zn complex with ethanol to generate a di-Zn complex (**Zn-3**) that improved *ee* up to 80% (Fig. 10b)[145].

Meanwhile, Coates and coworkers reported an imine-oxazoline Zn complex (**Zn-4**) for enantioselective ROCOP of CHO/CO₂ to produce PCHC with 76% *ee*[146]. Later, the same group identified an asymmetrical β-diiminate Zn complex (**Zn-5**) that enables the preparation of highly isotactic PCHC with units up to 92%[147]. Other examples are using chiral Zn catalysts: the amido-oxazoline

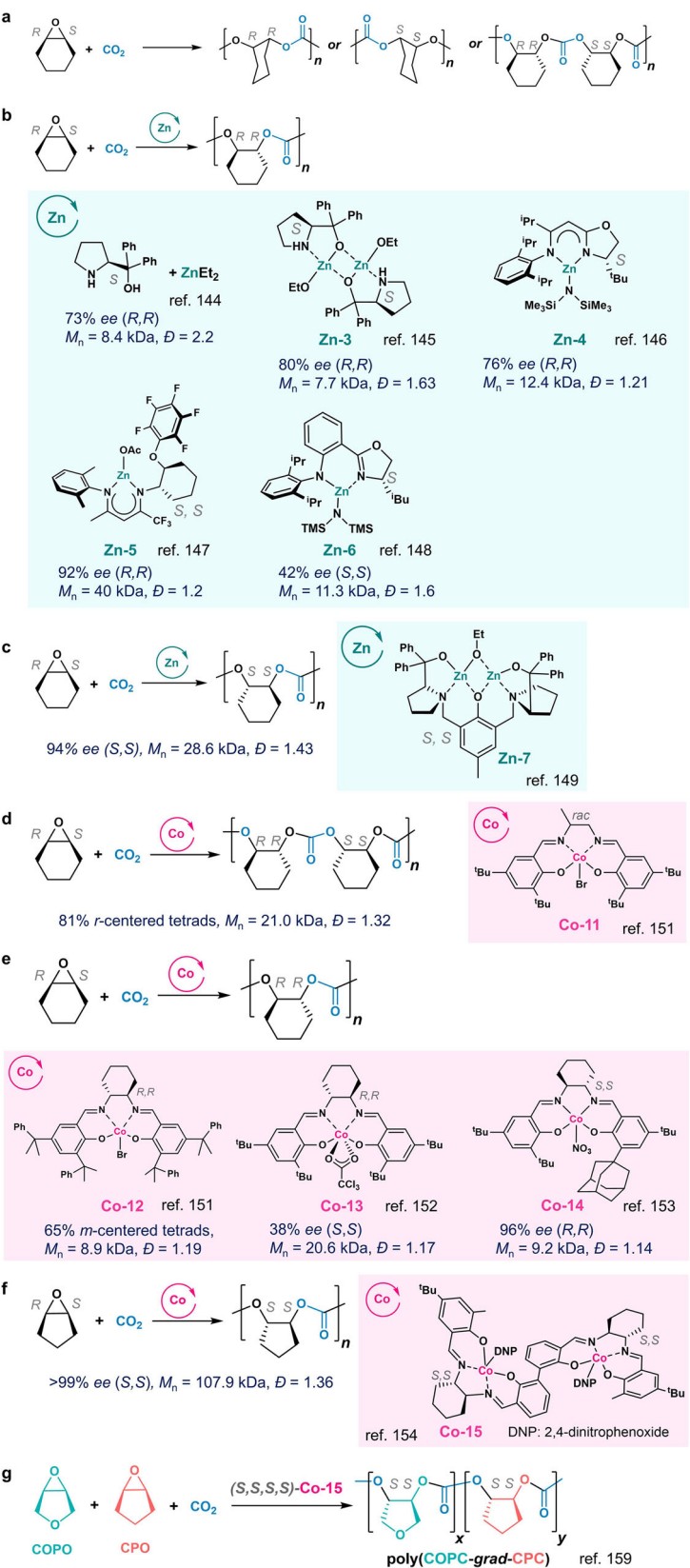

**Fig. 10 Enantioselective ring-opening copolymerization (ROCOP) of *meso*-epoxide/CO₂. a** Stereoselective ROCOP of *meso*-epoxide (e.g., cyclohexene oxide (CHO))/CO₂ to produce various stereoregular polycarbonates. **b** Zn-based complexes for isoselective ROCOP of CHO/CO₂. **c** bi-Zn complex for stereoselective ROCOP of CHO/CO₂ to produce (*S,S*)-enriched polycarbonates. **d** Co-11-mediated syndioselective ROCOP of CHO/CO₂. **e** Co-based complexes for isoselective ROCOP of CHO/CO₂. **f** Co-15-mediated isospecific ROCOP of CHO/CO₂ to produce (*S,S*)-enriched polycarbonates. **g** Co-15-mediated isoselective copolymerization of two *meso*-epoxides with CO₂ to produce (*S,S*)-enriched gradient polycarbonates.

Zn complexes (**Zn-6**) producing iso-enriched PCHC with units of 42% *ee* reported by Du and coworkers (all above in Fig. 10b)[148], and bi-Zn complexes (**Zn-7**)[149,150] that have (*S,S*)-enriched PCHC with 94% *ee* (Fig. 10c).

In 2006, the Coates group first reported syndioselective ROCOP of CHO/$CO_2$ using racemic (salen)$Co^{III}$ catalyst **Co-11**, which provided PCHC with 81% *r*-centered tetrads (Fig. 10d)[151]. Notably, adding [PPN]Cl as a cocatalyst decreased syndioselectivity. On the other hand, both Coates and Lu groups found that (*R,R*)-(salcy)$Co^{III}$ catalysts with bulky substituents (**Co-12** and **Co-13**) could produce iso-enriched PCHC[151,152]. In 2012, Lu identified an asymmetrical (salcy)$Co^{III}$ catalyst with the bulky *ortho*-substituent (**Co-14**) for enantioselective ROCOP of CHO/$CO_2$ to produce isotactic PCHC with 96% *ee* at −25 °C (Fig. 10e). The use of (*S*)-2-methyltetrahydrofuran as a chiral induction agent was found impactful for the enantioselectivity, whereas the inclusion of [PPN]Cl only slightly lower the *ee* to 90% at 0 °C[153].

To improve Co-catalyst's reactivity for stereoregular high-MW polycarbonate synthesis, in 2013, the Lu group developed bi-Co complexes in which two (salen)$Co^{III}$ moieties were linked through a biphenol linker. Combined with the cocatalyst [PPN][DNP] (DNP, 2,4-dinitrophenoxide), the enantiopure bi-Co complex **Co-15** could improve the reactivity and increase the selectivity over 99% *ee* in poly(cyclopentene carbonate) at room temperature (Fig. 10f)[154]. A bimetallic cooperation mechanism was proposed for the extraordinary enantioselectivity: the initiation is triggered by one of the two nucleophilic anions from the inside cleft of the bimetallic catalyst, and enchainment proceeded via the nucleophilic attack of the propagating carboxylate moiety at one Co toward the epoxide bonded at the other Co center. Both the stereochemistry of the biphenol-linker and cyclohexyl diamine skeletons determine the enantiomeric preference of the asymmetric copolymerization[155]. Such highly reactive and enantioselective chiral bi-Co complex could polymerize various epoxides with $CO_2$, producing the corresponding polycarbonates with more than 99% carbonate units and 99% *ee*[156]. The produced polycarbonates were mostly crystalline with $T_m$ values of 179–257 °C. The stereocomplex of these isotactic enantiopure polycarbonates prepared using **Co-15** exhibited significant improvement in $T_m$s and thermal stability[157,158]. Enantioselective terpolymerization of 3,4-epoxytetrahydrofuran, cyclopentene oxide, and $CO_2$ could be achieved by using the (*S,S,S,S*)-**Co-15** complex to produce gradient block copolymers with over 95% (*S,S*)-configured units, in which crystalline epoxytetrahydrofuran carbonate segment was incorporated first, followed by the amorphous segments consisting of cyclopentene carbonate (Fig. 10g)[159].

### Enantioselective polymerization of epoxide/anhydride

Aliphatic polyesters are commonly made through ROP of lactones[26,28,40,160]. The resulting polyesters have a limited range of properties owing to the lack of functional diversity of available lactones[161–163]. An alternative chain-growth route to both aliphatic and semi-aromatic polyesters is the alternating copolymerization of epoxides and cyclic anhydrides, which has emerged in recent 10 years and has been reviewed elsewhere[12,31,129,132]. Using two distinct monomer sets in such alternating ROCOP could allow for facile tuning of polyesters properties.

In 2016, the Lu group first reported enantioselective ROP of CHO (a *meso*-epoxide) with phthalic anhydride (PA) mediated by the enantiopure (*R,R,R,R*)-bi-Al complex **Al-11** with [PPN]Cl at 0 °C to afford (*R,R*)-enriched poly(cyclopentene phthalate) with 91% *ee* (Fig. 11a)[164]. Later, the same group improved the reactivity and enantioselectivity of bi-Al complexes by using a

hydrogenated binaphthol linkage; and the $^iPr$ group was found to be the optimal phenolate *ortho*-substituent for the bi-Al complex (**Al-12**, Fig. 11a)[165]. The obtained chiral polyesters are usually semicrystalline materials with $T_m$s up to 240 °C, and the obtained stereocomplex polyesters exhibited elevated $T_m$s of ~ 20-45 °C relative to the $T_m$s of the corresponding enantiopure counterparts[165,166]. The Lu group also identified another chiral bi-Al complex, **Al-13**, with the co-catalyst [PPN]Cl to mediate enantioselective ROCOP of racemic *cis*-internal *meso*-epoxide (e.g., 7-oxabicyclo[4,1,0]hept-2-ene) /PA at 0 °C to produce (*R,R*)-enriched copolyester with a high *s* factor of 437 (Fig. 11b)[166]. Very recently, Lu and coworkers identified a chiral bi-Cr complex **Cr-7**, along with [PPN]Cl, to mediate enantioselective ROCOP of racemic disubstituted *cis*-epoxides with cyclic anhydrides to produce (*R,R*)-enriched polyester with 90% *ee* and an *s* factor of 31 (Fig. 11c)[167].

In addition to *meso*-epoxides, the Lu group also applied the bi-Al complex for the enantioselective ROP of racemic mono-substituted epoxide and anhydrides. The chiral bi-Al complex **Al-14** having a binaphthol linkage, in conjunction with [PPN]Cl at 0 °C, exhibited exceptional enantioselectivities for the ROCOP of phenyl glycidyl ether (PGE)/PA with an *s* factor over 300 and over 99% *ee* (Fig. 11d)[168]. Such bi-Al complex also can be used for ROCOPs of various racemic epoxide/anhydrides, and all afford highly isotactic copolymers, with over 90% *ee* for most combinations. Additionally, the Lu group found that the combination of (*S,S,S,S,S*)-**Al-15** and [PPN]Cl could enantioselectively terpolymerize *rac-tert*-butyl glycidyl ether, *meso*-cyclopentene oxide, and PA to form a gradient copolymer with a 97% *ee* for the (*S,S*)-configuration in cyclopentene oxide units and a 95% *ee* for the glycidyl ether unit (Fig. 11e)[169]. During the enchainment, the (*R*)-*tert*-butyl glycidyl ether was incorporated first, followed by a gradually increased ratio of (*S,S*)-cyclopentene oxide, resulting in the copolymers bearing a wide range of $T_g/T_m$ temperatures by adjusting the co-monomer ratios.

Lu and Coates et al. also found that the stereocomplex assembly of discrete (*R*)- and (*S*)-aromatic polyesters required a minimal degree of polymerization over 5 and over 74% *ee*[170]. In addition, for aliphatic polyesters, the Coates group recently showed that only specific chiral combinations of epoxide and anhydrides would lead to the formation of semicrystalline stereocomplex mixtures, which could significantly increase $T_m$s[17].

### Other enantioselective ring-opening polymerization and copolymerization

**Carbonyl sulfide/*meso*-epoxide**. In 2018, the Lu group first achieved enantioselective synthesis of highly stereoregular poly(monothiocarbonate)s through the asymmetric copolymerization of *meso*-epoxides and COS (carbonyl sulfide) mediated by enantiopure biaryl-linked bi-$Co^{III}$ complexes (**Co-16**)[171]. A binaphthol-linked, bimetallic complex with an (*R,R,R,R*)-**Co-16** exhibited efficient enantioselectivity in mediating this asymmetric transformation, affording a variety of isotactic poly(-monothiocarbonate)s with over 99% isotacticity (Fig. 12a). Notably, the resultant chiral, sulfur-containing copolymers exhibit good optical properties with refractive indices of up to 1.56.

**Thiirane**. Early work on enantioselective ROP of methylthiirane showed low enantioselectivities[172–175]. In 1981, Spassky and coworkers found the chiral initiator of $ZnEt_2$/(*S*)-BINOL could selectively polymerize (*S*)-methylthiirane from the racemic mixture (Fig. 12b)[176,177]. Kakuchi and coworkers used $ZnEt_2$/L-leucine for the ROP of racemic (phenoxymethyl)thiirane, and found the *S*-enantiomer was preferably polymerized with an *s* factor of 5.36 (Fig. 12b)[178].

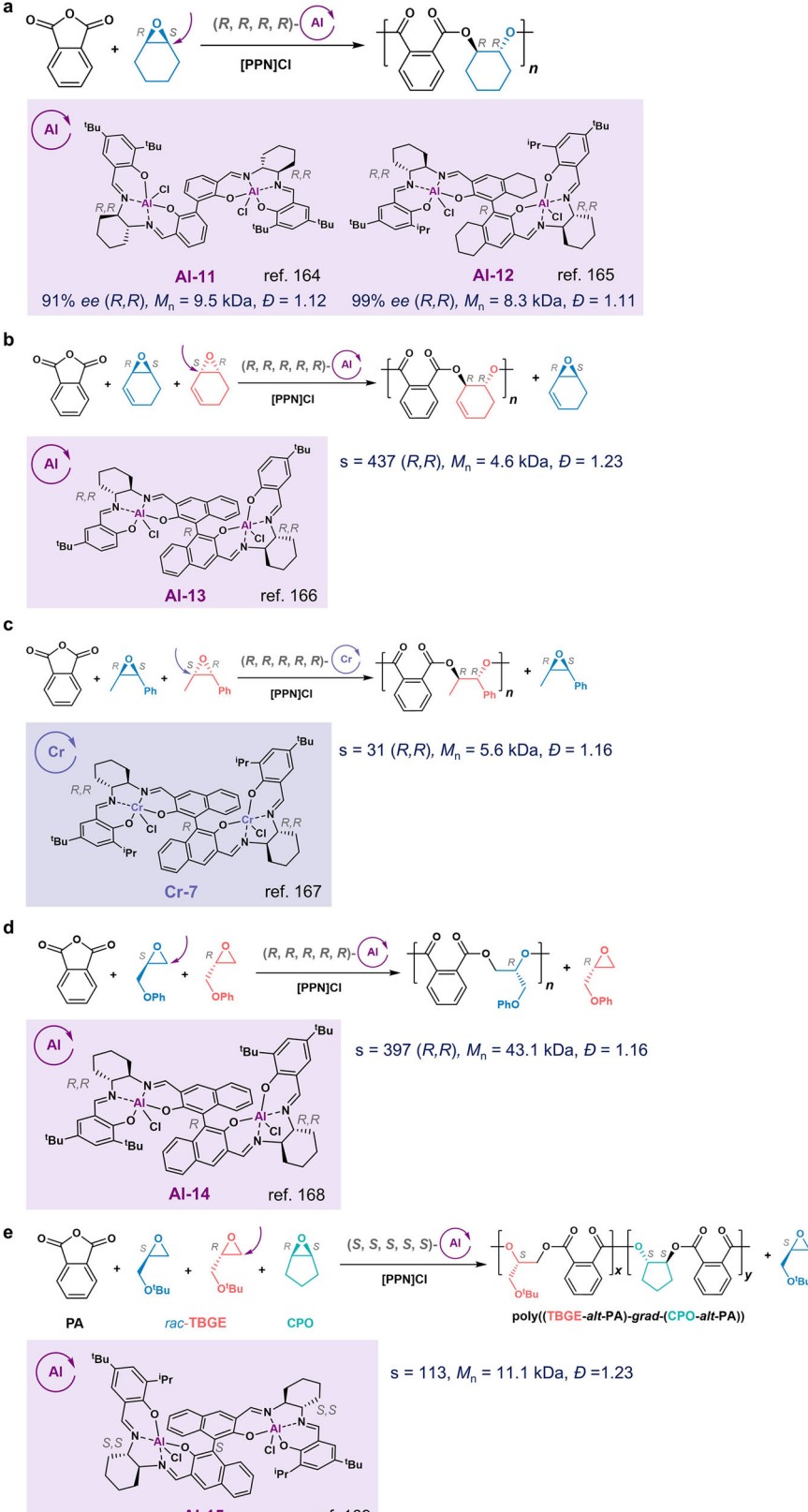

**Fig. 11 Enantioselective ring-opening copolymerization (ROCOP) of epoxide/anhydride. a** Al complexes for stereoselective ROCOP of *meso*-epoxide/ anhydride. **b** Al-13 and **c** Cr-7 mediated enantioselective ROCOP of *meso*-epoxides/anhydride. **d** Al-14-mediated enantioselective ROCOP of racemic mono-substituted epoxides/anhydride. **e** Al-15-mediated stereoselective ROCOP of *meso*-epoxides and racemic mono-substituted epoxides and anhydride.

**Fig. 12 Other enantioselective ring-opening polymerization (ROP) and ring-opening copolymerization (ROCOP) reactions. a** Enantioselective ROCOP of *meso*-epoxide/COS. **b** Enantioselective ROP of racemic thiirane. **c** Enantioselective ROP of racemic *N*-carboxyanhydride.

***N*-carboxyanhydride**. Synthetic polypeptides mimicking natural proteins offer potential advantages for biomedical applications with simpler components. Synthetic polypeptides can be prepared from enantiopure α-amino-acid *N*-carboxyanhydrides (NCAs)[179–181]. Early efforts included using chiral amines[182], AlEt₃/(+)-borneol[183], or 2-methylbutyrate-tri-*n*-butylphosphine Ni complex, etc.[184], but the overall enantioselectivities were very low. In 1999, Cheng and Deming found that Ni complexes with chiral 2-pyridinyloxazoline ligands could initiate ROP with a modest enantioselectivity (the $k_{rel}$ of 5.2, Fig. 12c)[185]. A higher $k_{rel}$ of 9.4 was achieved by the same group using a Ru complex with a dmpe ligand (dmpe, 1,2-bis(dimethylphosphino)ethane; Fig. 12c)[186].

## Thermal and mechanical properties of stereoregular polymers prepared via ROP

In Table 1, we summarize a few of the common stereoregular degradable polymers' thermo-mechanical properties. In general, high stereoregularity in the polymer backbone can lead to enhanced properties. In particular, the formation of stereocomplex—the mixture of two opposite enantiomeric polymers in equivalent amounts through an interlocked orderly packing of polymer chains—often results in a higher level of crystallinity and an increased $T_m$. For example, stereocomplex PLA exhibits a $T_m$ of 230 °C, which is ~ 50 °C higher than that of its enantiopure parent polymers[16]. Additionally, the stereocomplex PLA significantly increases its hydrolytic and thermal degradation resistance and gas barrier properties[187]. Coates, Lu, and coworkers showed that the weak hydrogen-bonding interaction between

carbonyl and methine of the opposite enantiomeric polyesters that were prepared by ROCOP of epoxides/anhydrides could be the driving force of stereocomplexation that leads to the greater increases in $T_m$s[170]. Similarly, the stereocomplex of poly(-limonene carbonate) was able to easily crystallize whereas its enantiopure parent polymers could not in spite of the high stereoregular configuration[188,189]. The ROCOP of racemic or meso epoxides with anhydrides using chiral Al complexes (Fig. 11a) could also lead to an increase in $T_m$ of ~ 25–45 °C for stereocomplex polyesters compared to the $T_m$ values of enantiopure constituents[165]. Coates, Tolman, and coworkers recently found that the specific chiral combinations between the epoxide and anhydride in the ROCOP could induce the change from amorphous to semicrystalline copolymers, and the stereocomplex of such copolymers exhibited increased $T_m$s[17].

Nevertheless, the high levels of stereoregularity and high crystallinity could bring up brittleness for the polymers, which limits the polymer's applications such as packaging. For instance, purely isotactic P3HB exhibits high crystallinity, high $T_m$ (~ 170–180 °C), high fracture strength (~35 MPa), and excellent barrier properties but is extremely brittle with low fracture strain (~5%)[74,87]. The introduction of stereo-defects into P3HB by using a less-stereoselective catalyst to produce syndiorich-P3HB (*sr*-P3HB) with a lower $T_m$ (114 °C) resulted in a significantly improved fracture strain (over 400%) without sacrificing the fracture strength (34 MPa)[190]. The change of stereoregularity or stereosequence renders a unique strategy to improve the polymer's properties without changing its chemical composition[163,190,191].

**Table 1 Thermo-mechanical properties of stereoregular polymers prepared via ROP[a].**

| Name | $P_m$ | $T_g$ (°C) | $T_m$ (°C) | $\sigma$ (MPa) | $\varepsilon$ (%) | Refs. |
|---|---|---|---|---|---|---|
| poly(L-LA) | 1 | 55–65 | 160–185 | 50–60 | 3–6 | 192 |
| poly(sc-LA) | 1 | 50–60 | 220–230 | 60–70 | 25–35 | 16 |
| poly(sb-LA) | 0.96 | – | 192 | 49 | 5 | 191 |
| poly(ht-LA) | 0.13 | 50 | – | 11 | 533 | 191 |
| poly(sc-3HB) | 0.99 | – | 171 | 35 | 3–5 | 87 |
| poly(sr-3HB) | 0.23 | ~0 | 114 | 34 | 419 | 190 |
| sb-PPO | 0.97 | – | 67 | 65 | 450 | 119 |
| it-PPO | 0.99 | – | 68 | 75 | 450 | 119 |

[a]$P_m$ the probability of meso linkages, $T_g$ glass transition temperature, $T_m$ melting temperature, $\sigma$ fracture strength, $\varepsilon$ fracture strain, sc stereocomplex, sb stereoblock, ht heterotactic, sr syndio-rich, it isotactic, LA lactide, 3HB 3-hydroxybutyric acid, PPO poly(propylene oxide).

## Conclusion and outlook

Enantioselective ROP and ROCOP of cyclic monomers have a long and distinguished history, with early work dating back to the early 20th century. Interest in the field has experienced a resurgence in recent years, due in part to the development of new synthetic methods and the urgency to find alternatives to non-sustainable petroleum chemicals. The application of NMR and chiral chromatography to elucidate polymer microstructures and reaction kinetics becomes critical to provide understanding of the polymerization process. Importantly, the polymerization catalyst dictates the reaction performance. Understanding the reaction mechanism, exemplified by Coates' work on enantioselective polymerization of epoxides[112–114], could aid the design of polymerization catalysts. Nevertheless, identification of qualitative trends in catalyst structures can be often challenging, because the catalyst structural factors influencing selectivity and polymerization efficiency in complex systems are high-dimensional. Notably, the machine learning techniques (e.g., Bayesian optimization) have been recently applied by our group to accelerate the discovery of stereoselective catalyst for the ROP of rac-LA mediated via the chain-end-control mechanism[191]. The development of such new methods for the quantitative identification of key structural features in catalysts will enable the unbiased optimization of catalyst performance and the discovery of new chemistry in this field. Additionally, as many studies focus on the structurally simple monomers, the expansion of monomers having pendant functional groups will help bestow new materials properties. Most reported polymerization systems have only been demonstrated on a bench scale (and many producing low-MW polymers), and thereby remain at a proof-of-concept stage. The examination of the catalyst and final polymer's toxicities is of importance if the polymers will be used for packaging or biomedical applications. The demonstration of materials performance and scalability of these new polymers, in addition to their degradability and sustainability, will make them attractive polymers for potential industrial translation in the future.

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

## Acknowledgements

This work was supported by the National Science Foundation (CHE-1807911).

## Author contributions

X.X., Z.H., and R.T. conceived the scope of the article. X.X., Z.H., E.J., and R.T. discussed the content and wrote the manuscript.

## Competing interests

The authors declare no competing interests.
