## [Peer Review File · Communications Chemistry]

Reviewers' comments:

Reviewer #1 (Remarks to the Author):

In this contribution, Tong et al. reviewed the recent advances in enantioselective ring-opening polymerization and copolymerization, which is a hot and significant research field. But for this review, there are the following problems to be solved.

1. First of all, the significance of this research direction is not clearly stated. Why it is necessary to realize the regulation of stereochemistry and microstructure of polymers, it is suggested to further elaborate in the introduction.
2. The biggest problem with this review is that it is highly coincident with the topics of the previous review (Coordination Chemistry Reviews 414 (2020) 213296), and the narrative perspective is also consistent. Compared with previous work, what is the advancement of this review? It is suggested that the author describe the novelty of this review, otherwise in my opinion, this review is meaningless.
3. The author should pay more attention to some details. For example, on page 3, what is "EMC-mediated enchainment"? Is it "ESC-mediated enchainment"? The writing of the rate constant should be unified, there are at least three writing types in this manuscript (k_{RR}/k_{SS} , $k(S,S)/k(R,R)$, $k(L-LA)/k(D-LA)$).
4. In addition, it would be of benefit to include the respective references in the figure. In Figure 5, different organic catalysts are suggested to be named for the convenience of readers.

Reviewer #2 (Remarks to the Author):

Stereochemistry is one of the main factors that determine the mechanical and physical properties of a polymeric material. The preparation of stereoregular polymers by stereoselective ring-opening polymerization of cyclic monomers has become one of the most promising methodologies in the field of polymer synthetic chemistry. This manuscript by Tong et al. reviewed in detail enantioselective ring-opening polymerization and ring-opening copolymerization of racemic monomers, which can produce microstructure-defined polymers.

The authors also provided a perspective on the future development direction. This review provides a comprehensive and detailed summary of the field and is very instructive for readers of the field.

Overall, the scholarly presentation of the article is very good, and the article presented logically and easy to read and follow. Thus, I recommend it for publication with the minor revisions suggested.

1. Some catalysts contain only one chiral center, it is inappropriate to name it as R,R or R,S, such as catalysts 1 and 3 in figure 5, Zn-4 in figure 10.
2. In figure 7h, the ROP of thiolactone by yttrium complexes is not enantioselective ROP although this polymerization produced isotactic polymer. It should be chain-end control mechanism by achiral catalyst.
3. It should be mentioned that artificial intelligence-assisted catalyst design for stereoselective ring-opening polymerization.

Reviewer #3 (Remarks to the Author):

The article by Tong et al. is a timely review that highlights recent advances in enantioselective ring-opening (co)polymerization. The review is well written and a good compilation of other reviews. Some minor points are listed below:

1. In the introduction, the authors state that they are only interested in enantioselective complexes, which have a remarkable potential for the preparation of new macromolecules. I partially agree with this statement. It would probably be appropriate to specify in paragraph 3 that certain complexes operating by a chain-end control mechanism are also quite capable of producing stereoblock PLAs with excellent activities (e.g. J. Am. Chem. Soc. 2004, 126, 2688–2689 ; Angew. Chem. Int. Ed. 2019, 58, 12585 –12589). In addition, it would be necessary to qualify the statement that chiral complexes are the most interesting for carrying out stereoselective polymerizations. Personally, I think the opposite: a catalyst (non-chiral) that works by a control mechanism at the end of the chain and that is also effective is more interesting because it is generally much less expensive.
2. In section 3.3, I find that some details about the systems used are missing. For example, the solvents used or the additives sometimes necessary are not always mentioned. In addition, it would be useful to specify that the toxicity of these derivatives is not necessarily lower than that of the metal derivatives.
3. In section 3.1, the Al catalysts described always require a temperature increase (to at least 70°C). It might be interesting to mention that the only example of an active (salen)Al complex in lactide polymerization is not stereoselective.
4. It would be preferable to have a table (or several) summarizing the thermal properties of the polymers (T_g, T_d and T_m). Some of them are mentioned in the text, but it is not systematic. However, the tacticity can have a significant influence on these properties.

Reviewers' comments:

Reviewer #1 (Remarks to the Author):

In this contribution, Tong et al. reviewed the recent advances in enantioselective ring-opening polymerization and copolymerization, which is a hot and significant research field. But for this review, there are the following problems to be solved.

1. First of all, the significance of this research direction is not clearly stated. Why it is necessary to realize the regulation of stereochemistry and microstructure of polymers, it is suggested to further elaborate in the introduction.

We thank the review's suggestion. We added "... stereoregular polymers, that is, polymers having regularity of the configurations of adjacent stereocenters along the polymer chain. Stereoregular polymers are usually crystalline and have improved thermal and mechanical properties relative to those of atactic polymers.¹"

2. The biggest problem with this review is that it is highly coincident with the topics of the previous review (Coordination Chemistry Reviews 414 (2020) 213296), and the narrative perspective is also consistent. Compared with previous work, what is the advancement of this review? It is suggested that the author describe the novelty of this review, otherwise in my opinion, this review is meaningless.

We acknowledge that our manuscript's topic overlapped with the Coordination Chemistry Reviews paper. We also added such review into the citation (ref. 13). We would like to out our manuscript having several different scope compared to the old one: (1) we discussed the enantioselective ROCOP of *meso*-epoxide/CO₂ (section 7.2), which was completely ignored in the 2020 review; (2) the ROCOPs of *meso*-epoxide/anhydrides and *meso*-epoxide/*rac*-epoxide/anhydrides were also discussed in our review (section 8), which were not mentioned in the 2020 review; (3) other emerging reactions including enantioselective ROP of β -lactones (Fig. 6c), ROP of diolides (Fig. 6h), ROP of β -thiolactones (Fig. 7h), ROCOP of COS/ *meso*-epoxide (Fig. 12a) have been discussed; (4) details of some enantioselective polymerizations were also discussed, e.g., the discussion of diolide ROP using different Y and La complexes for different stereosequences (Fig. 6d-g); whereas in 2020 review only 1 complex was mentioned even discussed on the same paper; (5) the detailed enantiomorphic mechanism discussion of PLA including recent computation results since 2019 was mentioned in our work (Fig. 3j) but was also ignored in the 2020 review. Furthermore, several details the nomenclatures—e.g., asymmetric kinetic resolution polymerization vs. enantioselective polymerization; we trying to use s factors at specific conversions not k_{rel} (which in many old literature papers refer to kinetic rates in the absence of the other enantiomer) for most reactions—are quite different, and we try hard to make the numbers useful for comparison. These significant advances mentioned in our review—but not appearing in the 2020 review—should be acknowledged instead of treated as “meaningless”, so do our efforts to make the structure, data, and literature appear in the same figures for reader's convenience.

3. The author should pay more attention to some details. For example, on page 3, what is “EMC-mediated enchainment”? Is it “ESC-mediated enchainment”? The writing of the rate constant should be unified, there are at least three writing types in this manuscript (k_{RR}/k_{SS} , $k(S,S)/k(R,R)$, $k(L-LA)/k(D-LA)$).

We thank for the reviewer's suggestion. We have changed all "EMC-mediated enchainment" to "ESC-mediated enchainment". We note that L-LA and D-LA are often used in the literature, and we changed all to terms such as $k(S,S)$ / $k(R,R)$ accordingly.

4. In addition, it would be of benefit to include the respective references in the figure. In Figure 5, different organic catalysts are suggested to be named for the convenience of readers.

We acknowledge the reviewer's suggestion, and references numbers were all added into Figure 3 to Figure 12. In addition, we named organocatalysts as **O-1**, **O-2**... in Figure 5 and corresponding discussions.

Reviewer #2 (Remarks to the Author):

Stereochemistry is one of the main factors that determine the mechanical and physical properties of a polymeric material. The preparation of stereoregular polymers by stereoselective ring-opening polymerization of cyclic monomers has become one of the most promising methodologies in the field of polymer synthetic chemistry. This manuscript by Tong et al. reviewed in detail enantioselective ring-opening polymerization and ring-opening copolymerization of racemic monomers, which can produce microstructure-defined polymers.

The authors also provided a perspective on the future development direction. This review provides a comprehensive and detailed summary of the field and is very instructive for readers of the field.

Overall, the scholarly presentation of the article is very good, and the article presented logically and easy to read and follow. Thus, I recommend it for publication with the minor revisions suggested.

We thank for the reviewer's comments.

1. Some catalysts contain only one chiral center, it is inappropriate to name it as R,R or R,S, such as catalysts 1 and 3 in figure 5, Zn-4 in figure 10.

We added "Note that $s = 4.4$ (S, S) means that the selectivity factor to (S, S)-LA is 4.4." in Figure 5 caption. We also labeled the chirality of the catalysts in Figure 5 and Figure 10.

2. In figure 7h, the ROP of thiolactone by yttrium complexes is not enantioselective ROP although this polymerization produced isotactic polymer. It should be chain-end control mechanism by achiral catalyst.

We acknowledge that the ligand is achiral, nevertheless the final Y complex is octahedral and has chirality. Indeed, we cannot determine whether such ROP was mediated via the chain-end control mechanism or enantiomorphic-site control. We noted "...the stereocomplex of cyclic polymers were proposed, though kinetic studies and the $ee\%$ measurement were not performed to eliminate the possibility of forming stereoblock microstructures via the chain-end control mechanism."

3. It should be mentioned that artificial intelligence-assisted catalyst design for stereoselective ring-opening polymerization.

In the conclusion section, we added “Notably, the machine learning techniques (e.g., Bayesian optimization) has been recently applied by our group to accelerate the discovery of stereoselective catalyst for the ROP of *rac*-LA mediated via the chain-end-control mechanism.¹⁸⁷”

Reviewer #3 (Remarks to the Author):

The article by Tong et al. is a timely review that highlights recent advances in enantioselective ring-opening (co)polymerization. The review is well written and a good compilation of other reviews. Some minor points are listed below:

We thank for the reviewer’s comments.

1. In the introduction, the authors state that they are only interested in enantioselective complexes, which have a remarkable potential for the preparation of new macromolecules. I partially agree with this statement. It would probably be appropriate to specify in paragraph 3 that certain complexes operating by a chain-end control mechanism are also quite capable of producing stereoblock PLAs with excellent activities (e.g. J. Am. Chem. Soc. 2004, 126, 2688–2689 ; Angew. Chem. Int. Ed. 2019, 58, 12585 –12589). In addition, it would be necessary to qualify the statement that chiral complexes are the most interesting for carrying out stereoselective polymerizations. Personally, I think the opposite: a catalyst (non-chiral) that works by a control mechanism at the end of the chain and that is also effective is more interesting because it is generally much less expensive.

We acknowledge that the papers that the reviewer mentioned and we cited them (refs. 41-42). Nevertheless, we have mentioned “Using different catalysts, the ROP of the racemic lactide mixture (*rac*-LA) of D-LA ((*R,R*)-LA) and L-LA ((*S,S*)-LA) can lead to isotactic stereoblock PLA or heterotactic PLA, whereas *meso*-LA ((*R,S*)-LA) can produce heterotactic or syndiotactic PLA.^{35, 39-42} Here we focus on the enantioselective ROP of LA.” We did not say chain-end control-mediated ROP of LA cannot produce useful materials but we would like to focus on enantioselective polymerization of *rac*-LA from a polymer chemistry point of view.

In addition, we emphasize using chiral complex for enantioselective polymerization. The use of achiral complex for chain-end control is not the review’s focus. Such mechanism is interesting in preparing stereoblock or other stereoregular polymers; nevertheless, the chain-end control mechanism cannot generate a stereocomplex PLA composed of two isotactic PLA (not stereoblock ones) or pure isotactic polymers. An example showing the importance of enantiomorphic control using chiral catalysts is the industrial production of isotactic polypropylene (*it*-PP) using the Natta catalyst or MAO catalysts. These chiral catalysts are inexpensive, interesting in chemistry, and applicable in industry. Again, the manuscript focuses on enantioselective polymerization and has never say that chain-end mechanism-mediate polymerization was inferior.

2. In section 3.3, I find that some details about the systems used are missing. For example, the solvents used or the additives sometimes necessary are not always mentioned. In addition, it would be useful to specify that the toxicity of these derivatives is not necessarily lower than that of the metal derivatives.

We thank the reviewer’s suggestion and add reaction conditions accordingly in section 3.3. As we focus on polymer chemistry and there are not many toxicology studies comparing metal and organocatalyst synthesized PLA, we would not make such statements about the materials or

catalysts' toxicities. We added "The examination of the catalyst and final polymer's toxicities is of importance if the polymers will be used for packaging or biomedical applications."

3. In section 3.1, the Al catalysts described always require a temperature increase (to at least 70°C). It might be interesting to mention that the only example of an active (salen)Al complex in lactide polymerization is not stereoselective.

We did not quite understand the reviewer's comments. There are many (salen)Al complexes not working well for stereoselective ROP. Also we focus on enantioselective ROP, and many (salen)Al complexes mediated ROP via the chain-end-control mechanism.

4. It would be preferable to have a table (or several) summarizing the thermal properties of the polymers (T_g, T_d and T_m). Some of them are mentioned in the text, but it is not systematic. However, the tacticity can have a significant influence on these properties.

We added Table 1 to summarize thermal and mechanical properties of some polymers mentioned in our manuscript.

REVIEWERS' COMMENTS:

Reviewer #1 (Remarks to the Author):

The author has solved the problems raised by the reviewers.

Reviewer #2 (Remarks to the Author):

After reviewing this revised version of the manuscript, I am happy that my concerns with the original manuscript have been addressed appropriately. Therefore, I am happy to support the publication of this manuscript.

Reviewer #3 (Remarks to the Author):

The authors have clearly taken criticisms and suggestions seriously and have made significant revisions to the manuscript. I am pleased that they have addressed some of my earlier comments. Furthermore, I believe these corrections should alleviate the concerns expressed by the other reviewers.